# IoT Off-Grid, Data Collection from a Machine Learning Classification Using UAV

**DOI:** 10.3390/s22197241

**Published:** 2022-09-24

**Authors:** Ademir Goulart, Alex Sandro Roschildt Pinto, Adão Boava, Kalinka R. L. J. Castelo Branco

**Affiliations:** 1Computer Science Graduate Program, Federal University of Santa Catarina, Florianópolis 88040-370, Brazil; 2Institute of Mathematics and Computer Science, Universidade de São Paulo, São Carlos 13566-590, Brazil

**Keywords:** IoT off-grid, data collection, drone, UAV, machine learning

## Abstract

IoT encompasses various objects, technologies, communication standards, sensors, actuators in powered environments, and networked communication. The concept adopted here, IoT off-grid, considers an environment without commercial electricity and commercial internet. Managing various utilities with IoT and collecting the relevant information from this environment is the purpose of this project. It uses machine learning to select relevant data. These data are collected safely using a drone that travels through the off-grid stations. A systematic literature mapping is presented, identifying the state of the art. The result is a software architecture proposal with configurations in the drone and off-grid stations that contemplate data collection from the IoT off-grid environment. The results are also presented with different selection algorithms used in machine learning and final execution in the prototype.

## 1. Introduction

IoT (Internet of Things) comprises a wide variety of objects and technologies, such as radio frequency identification (RFID), near-field communication (NFC), and wireless networks linking sensors and actuators. Furthermore, IoT encompasses different communication standards, protocols, and data formats, so the IoT environment is heterogeneous, decentralized, and complex. A fundamental component in this context is the sensors [1].

A sensor is a small, lightweight device that measures physical parameters such as temperature, pressure, relative humidity, and others in the environment. Sensor networks are highly distributed networks of sensor nodes deployed in large numbers to monitor an environment, or a system [2].

The concept considered here as IoT off-grid represents an environment in which we will have the installation of sensors and controllers in a place devoid of electricity and communication. Therefore, it is necessary to generate electricity to supply the site. This generation can be obtained with the use of photovoltaic equipment, wind energy, or another form of own generation of electric energy. A photovoltaic system with batteries sized for the load needed by the environment and an inverter will make it possible to meet the energy demand for the IoT off-grid environment. The components of an off-grid station are shown in Figure 1.

Initially, classification of the data produced in this IoT off-grid environment is performed so that only the selected data will be available to be collected. Using selection algorithms implemented via machine learning, data that meet the classification criteria will be eligible for future transfer.

The information thus selected will be collected periodically through a data mule (DM) [3], in this case, a drone, also identified in the literature as a UAV.

The contributions of this paper are the architecture described with its detailed configuration using selection algorithms that contributes to optimization in data collection and for the traveling salesman algorithm, in addition to considering the shortest path, it also considers the priority in collecting data at each point.

To the best of our knowledge, the information contained herein is accurate.

### 1.1. Existing Problems

In the off-grid environment considered here, data collection using a drone has the limitation of flight autonomy. A problem will be determining the shortest path to be traveled by the drone to make the most significant amount of collections at the points and return to base.

The classification and selection of primary data are necessary to reduce the volume of data to be transferred and take only the crucial data to be collected.

Another communication security problem is the need for the data transmitted between the base and the data collector to be encrypted.

During the communication process, a secure protocol must be used to avoid violations during the communication.

The lack of conventional power and communication infrastructure will be a problem to be solved at each station with a local power generation and communication solution.

Additionally, a problem related to the frequency of data collection must be addressed. Data collection points that have the oldest information should be collected first.

The equipment coupled with the drone must have sufficient storage capacity to collect data from different points. A battery and equipment must have the lowest possible weight to be transported by the drone.

### 1.2. Article Organization

In Section 2, the basic concepts are presented considering the topics of machine learning and classification; data collection; types of drones.

The correlated works are presented in Section 3, the result of a systematic literature mapping (SLM) in which the search was performed in eight databases, initially identifying 583 works and, in the end, selecting 20 works that met the defined criteria.

Section 4 describes the relevant aspects of this project considering the energy consumption, the security of the data communication between the drone and the collection point, and the storage capacity in the micro concentrator of the data at the collection point.

Possible solutions are given in Section 5 considering power generation with solar panels, the selection, and classification of primary data, a local server using Linux, which also behaves as an access point, and IEEE 802.11 communication with the equipment attached to the drone. The search path to access the points will be optimally based on the traveling salesman algorithm.

The project, the development of the proposal, and the results obtained are presented in Section 6, detailing the selection of data using machine learning, the configuration of the equipment coupled to the drone, and each data collection point. In the described prototype, three distinct points were considered. A complete script of the settings and programs involved, including security in data transfer, is available on GitHub https://github.com/ademirgoulart/IoT-Off-Grid-data-collection (accessed on 12 September 2022).

Final considerations are presented in Section 7.

## 2. Basic Concepts Involved

### 2.1. Machine Learning and Classification

Machine learning (ML) is a broad field that draws on concepts from computer science, statistics, cognitive science, engineering, optimization theory, and many other disciplines in mathematics and science [4]. There are numerous applications for machine learning, but data mining is the most significant of them [5]. Machine learning can be mainly classified into two broad categories, which include supervised and unsupervised machine learning.

Unsupervised machine learning is used to conclude data sets that consist of input data with no answers labeled [6]. The desired unsupervised learning output is not provided. Supervised machine learning techniques try to discover the relationship between input attributes (independent variables) and a target attribute (dependent variable) [7]. Supervised techniques can be further classified into two main categories: classification and regression. In the regression output variable, it takes continuous values, while in the classification output variable, it takes class labels [8].

Classification is a data mining (machine learning) approach to predict group membership for a data instance [9]. Although various techniques are available for machine learning, classification is the most commonly used technique [10].

Classification is categorized as one of the fundamental problems researchers study in machine learning and data mining. A general model of supervised learning (classification techniques) is shown in Figure 2 [11].

In the IoT off-grid environment, it is considered that a large volume of data is collected from locally installed sensors. To select which data collected from the sensors are most relevant, machine learning uses a previously installed classification algorithm separating the collected data by level of importance. Thus, according to the classification model, only the most critical data collected will be recorded in an eligible file and directory for transfer to the drone.

### 2.2. Data Collection

The data generated in this IoT off-grid environment will be collected, classified by an ML algorithm [9], and stored on a small local server. This local server will have limited storage capacity so periodical data collection will be necessary. Sending these data to a data center requires a specific solution, considering the lack of local communications infrastructure. In this project, the technique of data collection using a drone will be adopted. Data transfer is foreseen through wireless communication between the server and the drone.

Collecting data using UAVs has the advantage of not relying on commercial internet infrastructure in the IoT off-grid environment. The disadvantage in collecting data using UAVs is the flight autonomy which is limited to 30 min in this work. Thus, multiple flights will be necessary for data collection depending on the distance to the points.

### 2.3. Types of Drones

Drones, also known as UAVs (unmanned aerial vehicles), can have different sizes and load capacities. Another critical feature is autonomy because, depending on the battery capacity, the equipment can have autonomy for hours or minutes to perform the displacement and collection of data from the IoT off-grid environment.

A history of the evolution of drones can be found in [12].

Over the last years, drones have been widely used and were originally designed for military applications. With technological advances, size and cost reductions have become ideal for civil applications [13].

In a comparison of different platforms for data collection involving the public grid, satellites, tethered balloons, airships, and drones, Ref. [14] presents the advantages and disadvantages of drones. As advantages: large coverage area; large capacity throughput; low energy consumption; low deployment cost; high monitoring flexibility. Disadvantages: high data collection latency; limited terminal access; susceptibility to weather conditions.

## 3. Related Works

A complete systematic literature mapping (SLM) related to this topic and produced by the authors was published in [15]. The following is a part of this mapping related to the current work to identify the state of the art.

### 3.1. SLM Research Objective and Research Question

The main objective of this study was to systematically map current research on data collection using a data mule in geographically dispersed IoT.

As research questions (RQs), we have:RQ1—What algorithm is used in routing for data collection?RQ2—What technology is used for the reception of data by the drone?

To successfully search for important studies, search terms are critical. The following search string was used.

STRING: IoT AND (UAV OR Drones) AND “data collection”

### 3.2. Selection Procedure

The selection procedure aims to identify significant studies for the SLM. The execution of the search in the different bases from the search string presented a primary result of 583 works. After all the selection procedures, a final list of the 20 selected works is presented in Table 1.

### 3.3. RQ1—Algorithms Used in Routing

A list of the algorithms used in routing is presented in Table 2.

### 3.4. RQ2—Technology Used for Data Reception

A list of the technologies used to receive the data, referring to 11 works, is presented in Table 3.

### 3.5. Relationship and Comparison with Related Works

The present work differs from the other correlates because it deals with an environment considered off-grid, that is, a place where there is no commercial electrical infrastructure or public communication. It deals with IoT in this environment by collecting data generated there that must be transported to an environment with public communication for their processing.

Data collection in each off-grid environment considered here considers the security of the information collected for the drone. Not only secure authentication in the wireless connection but also the encryption during the transmission from the station to the micro collector in the drone are contemplated in this work.

The data transferred from the off-grid environment is pre-selected using machine learning techniques with classification algorithms aiming at energy savings. Only the necessary data according to the algorithm’s prediction are transferred to the drone.

## 4. Relevant Aspects

Considering that the off-grid environment does not have commercial electrical energy but some form of on-site generation, the energy generation capacity will be minimal enough to supply the electronic devices installed in this environment. Sensors, actuators, local servers, and data transmission equipment must be dimensioned for the lowest energy consumption. Communication between the local server and the data collection equipment must be secure, using secure data communication and encryption protocols, ensuring the integrity of the data transferred to the collecting equipment.

Aiming at lower consumption and greater efficiency in integrating sensor, server, and data transmission equipment, a single device is planned to perform all these functions in an integrated manner.

The collected data will be stored on a physical device of limited capacity. Thus, these data must be collected at time intervals compatible with this storage capacity, avoiding data loss due to a lack of storage space.

As examples of use for an off-grid environment, we could mention data collection in agricultural machines [26], data collection in meteorological stations [16], data collection in remote utilities managed by sensors [18], data collection in isolated environments [14], and other equivalent characteristics of an off-grid environment.

## 5. Possible Solutions

With solar panels or mechanical power generation, adequate batteries, and a local server, there are conditions for implementing sensors and actuators to control various utilities, based on IoT, in an off-grid environment.

The selection and classification of primary data will be performed using machine learning classification algorithms.

The data generated in this IoT off-grid environment will be collected and stored on a small local server using the Linux operating system. Sending these data to a data center requires a specific solution, considering the lack of local communications infrastructure. In this project, the technique of data collection through a Drone using IEEE 802.11 communication will be adopted.

Safety in the off-grid environment is considered in wireless communication and during data transfer to the drone. Wireless communication uses WPA2, which employs the CCMP (cipher block chaining message authentication code protocol), which uses a very robust and widely used symmetric encryption algorithm, AES. The SCP protocol that uses SSH is used for secure data transfer.

For data collection at each point of the off-grid environment in this project, the use of a drone as a data mule is foreseen. A small board with storage and wireless communication and a battery to power this board will be attached body of the drone.

At each point of the off-grid environment, the information coming from the sensors will be stored in a small local server with adequate storage capacity. This same local server will also be an access point for wireless communication.

Considering the drone’s departure from its point of origin and searching for information in different IoT off-grid environments, it is necessary to define a search path to go through all the different locations. For that, an optimized algorithm must be adopted that is similar to the traveling salesman algorithm, but with several restrictions beyond the physical distance, such as drone flight autonomy and priority depending on the age of the data.

In this project, we will focus on maintaining the flow of information, more specifically on the problem known as data mule scheduling (DMS). In this case, the robot in question, a drone, is considered a data mule (data mule—DM) and must traverse the IoT off-grid networks deployed in a given area to collect data.

An architecture diagram for IoT off-grid and drone data collection is presented in Figure 3.

Consider the total area that will be monitored taking into account the autonomy of the drone, which will be a critical point in the project. In a complete scenario, the types of sensors to be adopted must be considered. The data to be collected must be classified with an efficient algorithm compatible with the type of data collected.

A point to consider is the health station that can be collected to monitor the station’s good functioning.

If it is possible to schedule data collection, the WiFi activation can also be activated only during the data collection period.

Environmental conditions relating to temperature and other possible local interference must also be considered.

## 6. Project, Development of a Proposal, and Results Obtained

To validate the proposed solution presented in this work, a scenario with three different locations where off-grid environments were configured and a drone coupled to a micro data collector was used, as shown in Figure 4. Algorithms for classifying the data collected by the drone using machine learning were tested, and the results re presented subsequently.

### 6.1. Data Selection Using Machine Learning

For the validation of machine learning using data from an IoT environment, a data set called “Environmental Sensor Telemetry Data” was obtained from the website kaggle [33]. In this data set, we find the following information collected in three different IoT devices: temperature, humidity, Co, LPG (liquid petroleum gas), smoke, light, and motion.

The total number of records is 405,184 from three data collection environments. The purpose of machine learning is to classify the critical data to be collected by the drone in each IoT station. In this context, data will be important when indicating the presence of a person in the environment. Thus, based on the various parameters in each reading, the classification algorithm predicts if there was the presence of any person in the environment.

Considering the energy optimization during the collection process, the classification performed by the algorithm reduces the number of records to be transferred to the drone, as shown in the results presented subsequently.

#### 6.1.1. Algorithm Accuracy Results

For the classification of records using machine learning, the following algorithms were evaluated: Logistic Regression [34,35]; Naive Bayes [35,36]; Decision Tree [37]; Random Forest [35]; KNN (K Nearest Neighbor) [35,38]; SVM (Support Vector Machine) [39,40]; Neural Network [35].

The best accuracy, 98.97%, was obtained by the Random Forest algorithm. Figure 5 shows the accuracy comparison among the algorithms.

The confusion matrix referring to the algorithms is shown in Figure 6.

#### 6.1.2. Algorithm Execution Results

In the learning phase of the algorithms, a ratio of 80% of the data was used for learning and 20% for testing. Different execution times were obtained using the Raspberry Pi, and the SVM algorithm was the most time-consuming according to Table 4.

In order to select only the classified records to be collected by the drone, an execution reading of the 405,184 records of the data set classified the number of records. For each algorithm, the number of classified records and the runtime are presented in Table 5.

#### 6.1.3. Communication with the Drone

The communication between each off-grid station and the drone occurs automatically after the connection between the drone and the station. The station is configured as a server and an access point using a WiFi connection.

The drone searches for the SSID of the station and when it finds it becomes a client of this access point. Once the connection is established, the selected data is transferred from the off-grid station to the drone. The drone remains stationary for a while, parameterized in the routing algorithm, and then goes to the next point.

In the tests performed with three points, the times required for the complete transfer of data are shown in Table 6. Times show the transfer of the original data, 405,184 records, and the selected data set, 111,815 records, after selection as the machine learning algorithm with the highest accuracy.

### 6.2. Configuration at Each Off-Grid Station

Different actuators and sensors are assumed to manage local utilities at each off-grid location. The information collected from the various sensors is sent to the computer responsible for collecting and storing the information locally. The algorithm developed with machine learning previously configured for the environment performs a data classification process. All classified data are stored in a single file for later transfer to the drone.

This same micro collection was configured as an access point, allowing a wireless connection and the transmission of the data initiated after the drone’s approach to the place.

Raspberry PI boards were used in the evaluated scenario. One model 3B and two model 4Bs. Each micro Raspberry, after its configuration as an access point, was also configured to accept connection via SSH using local and remote encryption keys, dispensing with the use of login and password for the connection between the server and the drone.

The transfer of files from the station to the micro attached to the drone is via SSH, and thus it is necessary to allow the use of SSH on port 22 on the station via Raspberry settings.

In Figure 7, we have one of the Raspberry PI boards used in the prototype validation.

### 6.3. Configuration on the Micro Attached to the Drone

In the evaluated scenario, a DJI AIR 2S Drone was used with a 5000 mAh power bank attached to its body, feeding a Raspberry Pi Zero 2W and responsible for collecting data at remote stations.

This Raspberry Pi Zero 2W coupled to the drone automatically detects the wireless network of each of the access points of the off-grid stations. To allow the drone’s Raspberry to identify the SSID of each off-grid station, it is necessary to configure WPA_SUPPLICANT. This component is used on client stations and implements key negotiation and authentication/association in roaming.

A Python script is started after the computer’s boot. This script in the computer coupled to the drone is responsible for identifying the off-grid station and copying the data. A copy of this script and details of other mentioned settings can be accessed at this GitHub https://github.com/ademirgoulart/IoT-Off-Grid-secure-data-collection-from-a-machine-learning-classification-usin-UAV (accessed on 12 September 2022).

In Figure 8, we have a Raspberry PI Zero 2W board attached to the drone that was used to validate the prototype.

### 6.4. Routing Algorithm for Data Collection

In the previously presented SLM, the TSP algorithm is identified as predominant for determining a route to be followed by the drone during the data collection activity at off-grid stations. Second [12], a taxonomy of representational techniques related to project planning paths in heuristic search techniques using the Greedy Algorithm or Genetic Algorithm.

In the present work, a technique for the solution of the TSP known as the Greedy heuristic search algorithm [12] meets up to 415 points according to tests performed. Execution times and numbers of different routes to meet the points are presented in Table 7.

An adapted version of the work in [28] to allow the calculation of multiple routes is presented in the following algorithm.

The definition of $S_{i}$ relates to the smallest distance and priority $Pr_{i}$. We calculate the normalized distance $D_{c}n,i$. The best point *i* is obtained by multiplying priority $P_{i}$ and dividing by normalized distance $D_{c}n,i$.
$$S_{i}=\frac{Pr_{i}}{D_{c}n,i}$$
$$D_{c}n,i=\frac{D_{c},i-min\left(D_{c},i\right)}{max\left(D_{c},i\right)-min\left(D_{c},i\right)}$$

The priority is an input to the algorithm. Another input data for each point is the X and Y coordinates. Thus, the shortest distance between two points is calculated from their coordinates. The $S_{i}$ index considers the smallest distance related to the priority at the point. Thus, after normalizing the distance, the $S_{i}$ calculation considers the priority divided by the smallest distance. The largest $S_{i}$ will be selected as it represents the point with the smallest distance related to the highest priority (Algorithm 1).
**Algorithm 1:**  Routing Algorithm for Data Collection
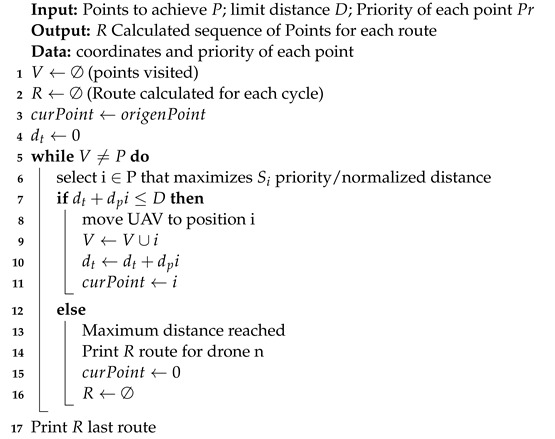


So multiple routes can be used so that all points are achieved by multiple UAVs or a single UAV performing the calculated routes sequentially. When calculating these multiple itineraries, the smallest distance to be covered and the search priority at each point is considered. The distances are normalized in advance, resulting in an interval between 0 and 1. This normalized distance is multiplied by the highest priority value to have an equivalent scale of measurements. With these normalized values, we calculated an index of the priority ratio of each point divided by the distance. So the highest index is used to determine from any point the next point be selected, considering the shortest distance and the highest priority.

The results obtained in a run to determine the path with 10 points without considering priority in each point are shown in Figure 9. Considering the drone’s autonomy, the first route is composed of points 0, 1, 2, 3, 4, 5, and 8 and returns to its origin. A second route completes the 10 points, is composed of points 0, 9, 10, 7, and 6, and returns to the origin.

The results obtained in a run to determine the path with 10 points considering the priority of each point, identified by the Pnn value, are presented in Figure 10. In this case, considering the autonomy of the drone, the first route is composed of points 0, 2P6, 3P5, 8P10, 9P9, 7P8, and 6P7 and returns to the origin. The second route to complete the 10 points is composed of points 0, 4P4, 5P2, 10P3, and 1P1 and returns to the origin.

## 7. Final Considerations

The solution presented in this work shows that it is possible to collect data in an off-grid environment using a drone that transports a small computer to collect information wirelessly.

With equipment such as the Raspberry Pi Zero 2W powered by a 5000 mAh battery, adding a total weight of 150 g to the drone, it was possible to collect data from different stations.

Concerning data communication security, this work considered the authentication between the station and the micro collector and the data encryption during the data transfer period.

The same station off-grid was configured in a Raspberry PI for simultaneous use as an access point, data server, and data selection based on machine learning.

A previously developed algorithm using machine learning, loaded on the off-grid station, selects relevant information using prediction techniques.

The selection of only priority records to be transferred to the drone contributes to energy savings in the data collection process.

As a future work, one could increase the number of collection points and use multiple drones. Using network simulation software, one could simulate wireless connection conditions by varying link distance and speed to determine error rate and connection thresholds.

## Figures and Tables

**Figure 1 sensors-22-07241-f001:**
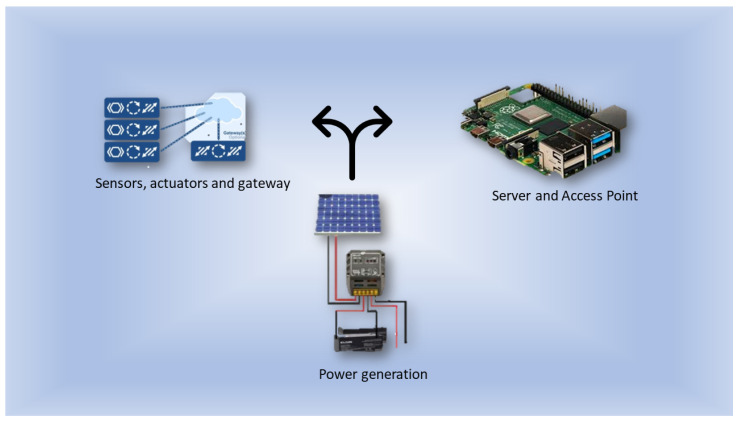
Off-grid station.

**Figure 2 sensors-22-07241-f002:**
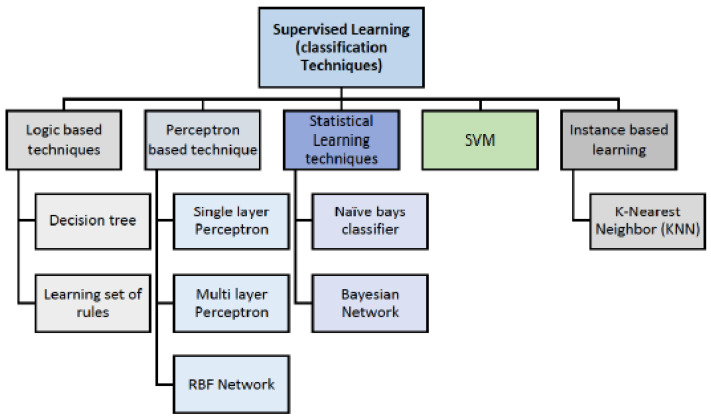
Supervised learning classification techniques [11].

**Figure 3 sensors-22-07241-f003:**
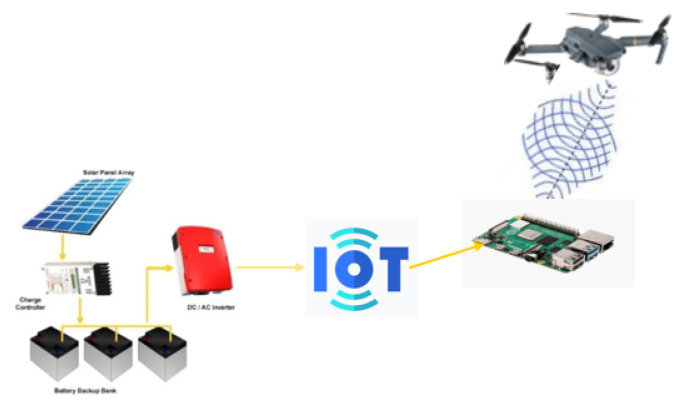
IoT off-grid architecture diagram.

**Figure 4 sensors-22-07241-f004:**
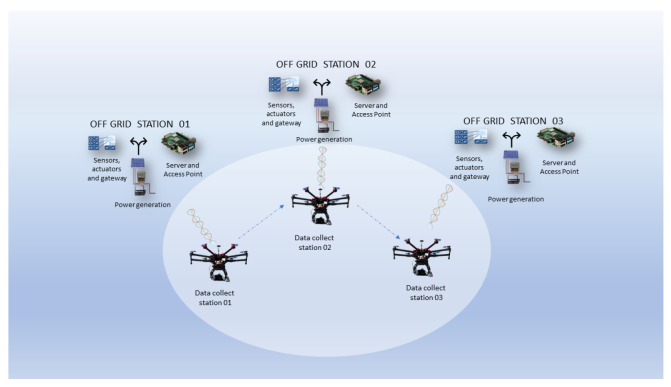
Scenario with off-grid stations and drone displacement.

**Figure 5 sensors-22-07241-f005:**
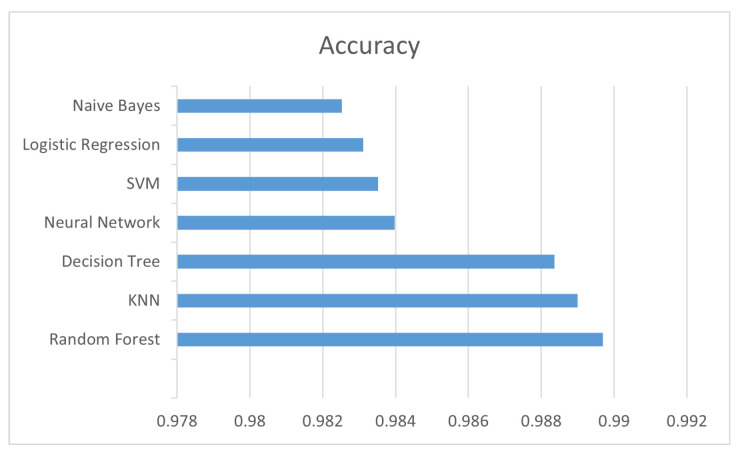
Accuracy.

**Figure 6 sensors-22-07241-f006:**
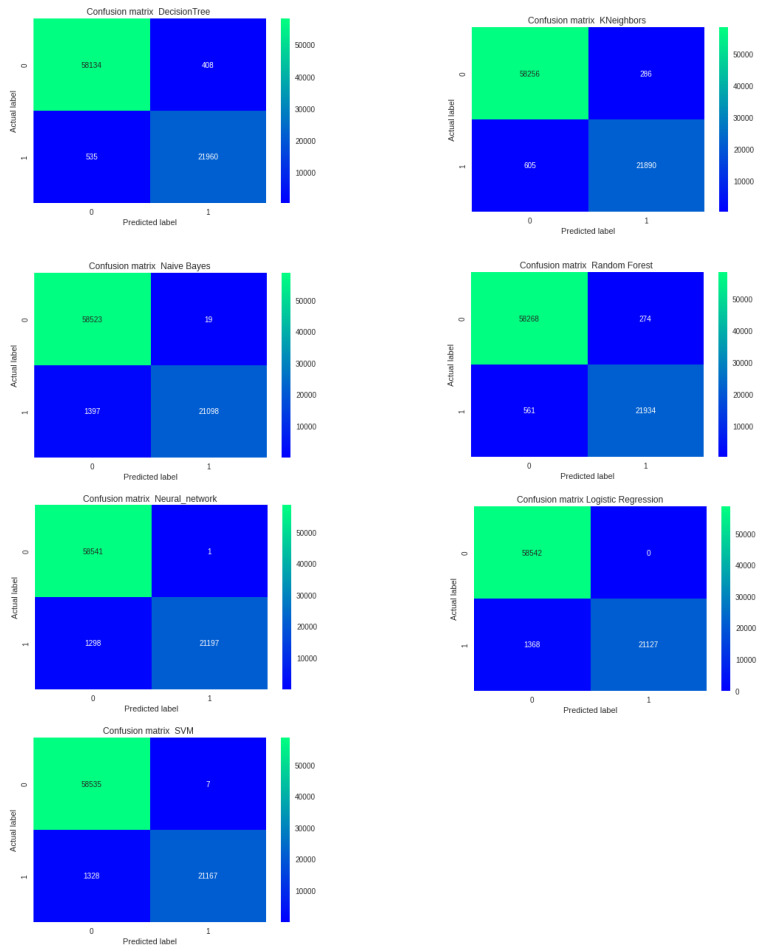
Confusion matrix referring to the algorithms.

**Figure 7 sensors-22-07241-f007:**
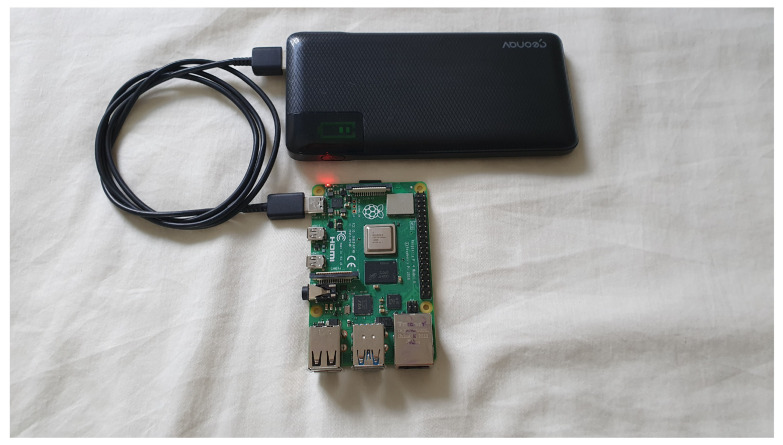
Off-grid station—Raspberry Pi.

**Figure 8 sensors-22-07241-f008:**
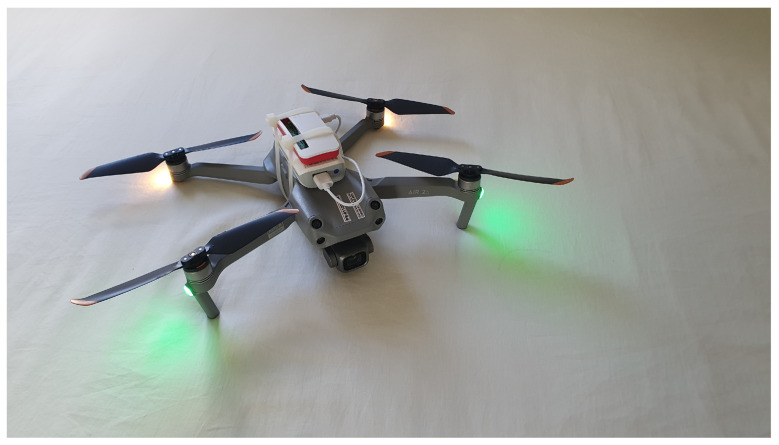
Drone attached to the Raspberry Pi Zero 2W.

**Figure 9 sensors-22-07241-f009:**
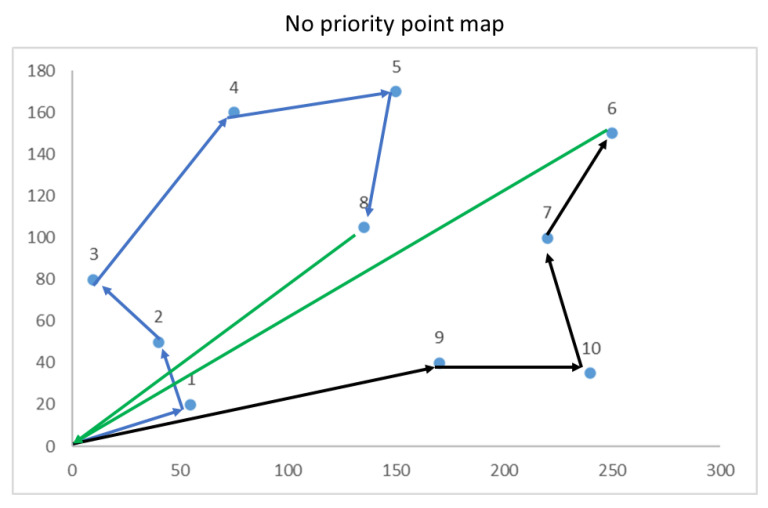
Route of points without priority (unit: meters).

**Figure 10 sensors-22-07241-f010:**
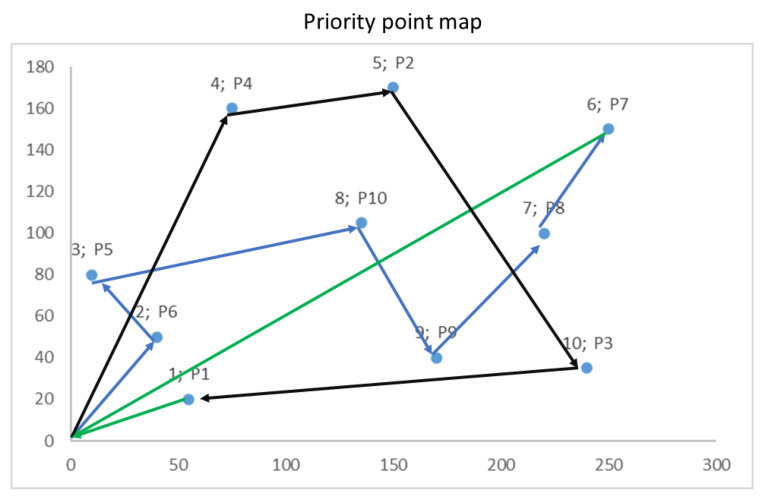
Priority points route (unit: meters).

**Table 1 sensors-22-07241-t001:** Selected papers.

Reference	Title
[16]	A new system for agrometereological data collection in areas lacking communication networks
[17]	A precision adjustable trajectory planning scheme for UAV-based data collection in IoTs
[18]	A solution for data collection of large-scale outdoor internet of things based on UAV and dynamic clustering
[19]	A Survey of Key Issues in UAV Data Collection in the Internet of Things
[20]	BEE-DRONES: Ultra low-power monitoring systems based on unmanned aerial vehicles and wake-up radio ground sensors
[21]	Data collection using unmanned aerial vehicles for Internet of Things platforms
[14]	Drone-Enabled Internet-of-Things Relay for Environmental Monitoring in Remote Areas Without Public Networks
[22]	Dynamic Rendezvous Node Estimation for Reliable Data Collection of a drone as a Mobile IoT Gateway
[23]	Efficient and Reliable Aerial Communication with Wireless Sensors
[24]	Efficient data collection by mobile sink to detect phenomena in internet of things
[25]	Environmental Monitoring Using a drone-Enabled Wireless Sensor Network
[13]	Internet of Things Data Collection Using Unmanned Aerial Vehicles in Infrastructure Free Environments
[26]	LoRa Communications as an Enabler for Internet of drones towards Large-Scale Livestock Monitoring in Rural Farms
[12]	Path planning techniques for unmanned aerial vehicles: A review, solutions, and challenges
[27]	Performance Evaluation of 802.11 IoT Devices for Data Collection in the Forest with drones
[28]	UAV path planning for emergency management in IoT
[29]	Area Division Cluster-based Algorithm for Data Collection over UAV Networks
[30]	A Brief Review of the Intelligent Algorithm for Traveling Salesman Problem in UAV Route Planning
[31]	Age-optimal trajectory planning for UAV-assisted data collection
[32]	Age-optimal path planning for finite-battery UAV-assisted data dissemination in IoT networks

**Table 2 sensors-22-07241-t002:** Algorithms used in routing.

Reference	Algorithm
[17]	*PATP—Precision adjustable trajectory planning*
[18]	*Ant colony algorithm*
[20]	*TSP-Ant colony optimization (ACO)*
[21]	*BL-TSP algorithm*
[24]	*Path based on the order of Hilbert values*
[13]	*Hilbert curve-based path planning algorithm*
[26]	*TSP and enhanced particle swarm optimization (EPSO)*
[28]	*Generalization of TSP*
[29]	*Simple area division cluster-based algorithm (SAD-CA)*
[31]	*Max-AoI-optimal and Ave-AoI-optimal*
[32]	*Stage-WSHP*

**Table 3 sensors-22-07241-t003:** Types of communication.

Reference	Communication Technology
[16]	*Bluetooth Low Energy (BLE)*
[18]	A *ZigBee wireless 2.4 GHz*
[20]	*Simple request/replay subGHz radio*
[21]	802.11b (no simulador)
[14]	*LoRa e IEEE 802.11 ac (5 ghz)*
[22]	Simulador com IEEE 802.15.4
[23]	*ContikiMAC over the IEEE 802.15.4 2.4 GHz*
[25]	*WiFi*
[13]	*Device with the DTN protocol implemented*
[26]	*Multi-channel LoRaWAN^®^ gateway*
[27]	*WiFi 802.11 2.4 GHz*

**Table 4 sensors-22-07241-t004:** Raspberry PI learning times.

Algorithm	Time (mm:ss, s)
Random Forest	00:44, 8
KNN	00:41, 9
Decision Tree	00:05, 0
Neural Network	12:29, 3
SVM	48:35, 7
Logistic Regression	00:10, 1
Naive Bayes	00:02, 7

**Table 5 sensors-22-07241-t005:** Raspberry PI final runtime and quantity.

Algorithm	# Classified	Runtime (mm:ss, s)
Random Forest	111,815	00:05, 8
KNN	111,012	00:59, 6
Decision Tree	111,566	00:01, 4
Neural Network	105,955	00:02, 6
SVM	105,859	31:14, 0
Logistic Regression	105,611	00:01, 3
Naive Bayes	105,558	00:01, 9

**Table 6 sensors-22-07241-t006:** Number of records and communication time.

Station	Time to Transfer 405,184 Records (Seconds)	Time to Transfer 111,815 Records (Seconds)
Station 1	23	5
Station 2	27	8
Station 3	19	5

**Table 7 sensors-22-07241-t007:** Number of points, runtime, and number of routes.

# Points	Runtime (Seconds)	# Routes
10	2, 23	2
50	3, 37	10
100	8, 45	19
200	28, 16	38
400	88, 80	73
415	97, 85	76

## Data Availability

Documentation of the procedures and programs used during the development of the article is available in the GitHub repository, which can be found at the URL https://github.com/ademirgoulart/IoT-Off-Grid-data-collection (accessed on 12 September 2022).

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
