# Peer review of "IoT Off-Grid, Data Collection from a Machine Learning Classification Using UAV"

_sensors, 2022, doi:10.3390/s22197241_

Round 1

Reviewer 1 Report

In this paper, the authors mainly discuss the data collection problem in IoT Off-Grid environment. Drones are used to traverse between Off-Grid stations to achieve the data collection. In this process, the machine learning classification algorithms are used to reduce the transmission of non-critical data, and an adapted search path planning algorithm is used to plan the UAV route, which realizes the collection of Off-Grid data and gives relevant experimental results. The work of this paper is meaningful and interesting. It is a useful complement to the applications of the IOT. However, the paper has the following problems:

1) Although the issues involved in the paper are interesting, the whole paper is not systematic. This paper mainly applies the existing classification algorithms and path planning algorithms to solve the data collection problem of Off-Grid stations. It does not make necessary improvements to the relevant algorithms, so its innovation is general. The author should clearly list the contributions of this paper in the introduction.

2) IoT Off-Grid environment is definitely different, but the characteristics of IoT Off-Grid are not particularly considered in the paper in terms of methods and algorithm design, such as WiFi, which is can be used both in IoT and IoT Off-Grid environments. The characteristics of IoT Off-Grid environments are not considered in such as classification algorithms and path planning algorithms. Please supplement and explain.

3) The paper mentions the security issue, how to ensure security in IoT Off-Grid environments, and how is it different from that in general IoT  environments?

4) The classification algorithms used are all ordinary machine learning algorithms. It is suggested to try some new algorithms, such as XGBoost, CNN, etc.

5) In section 6.4 routing algorithm, two parameters smallest distance and priority are mentioned. Please explain the meaning of smallest distance and how the priority Pri is determined? How Si is used, please supplement the specific description of Si.

6) I suggest that the author give a more detailed system architecture diagram or overall solution diagram in section 5 or section6 of the paper, so as to better understand your paper

Reviewer 2 Report

The purpose of this paper is to manage various public utilities through the Internet of things and collect relevant information from this environment. It uses machine learning to filter relevant data in the network. The UAV collects these data safely through the off-site site. The algorithm of UAV routing is analyzed and selected. A systematic document mapping is proposed and the latest technology is identified. The result is a software architecture proposal, which includes the configuration of UAVs and off-line websites, and considers collecting data from the off-line environment of the Internet of things. Different selection algorithms are used in the machine learning and final execution of the prototype, and the results are given. In addition, it has a wider range of applications.

However, there are some related questions:

1.     In Section 2 of Chapter 2, the article uses UAVs to collect data in the off-line environment of the Internet of things, but does not compare the advantages and disadvantages of the method of collecting data using UAVs and the non UAV data collection method, that is, using specific communication solutions.

2.     In Section 6 of Chapter 1,  Although the article compares the advantages and disadvantages of centralized machine learning methods, and compares the advantages and disadvantages in training time and accuracy, the final scheme is based on the method with the highest accuracy. What is the significance of comparing the above machine learning methods from different angles? What is the reason for choosing this method only through a single performance? It also did not explain which algorithm was selected for data collection

3.     The format of the article is not standardized, for example:The beginning letter of the title in ‘Figure 6. confusion matrix referring to the algorithms is lowercase’, which is inconsistent with the previous text. In addition, the clarity of some block diagrams does not seem to be very high, for example, figure 2 has obvious difference from other pictures

4.     The workload of this paper seems to be relatively rich, but the description of innovation points seems not to be particularly clear. It is necessary to further summarize the work done and highlight the innovation points.

Reviewer 3 Report

Authors describe a solution to collect sensor data in environment with no connection and no electricity. They select ML algorithm running on stations  in order to find important data items to send to drone.

The related works and the algorithms comparison are fine and well described.

The validation scenario is sufficient to show the idea could be implemented and it is just the starting point. In a similar case, two sections should be added to article:

- a user scenario to drive the development of the project and overcome the limits of the validation scenario (sensors, area to monitor, data to collect, objectives, ...)

- a section with next steps and future works

Authors do not consider to collect health-station data with the drone. The station is a sensor data collector and a sensor at the same time. Station will be placed in hostile areas with no electricity and connection and know their health will help to know when they must repaired.

Authors could consider to data collection by the drone could be scheduled and stations could switch-on wifi on prefixed times. It will reduce the battery pack to install in stations.

Sensors, electricity generation and batteries of stations are not listed or considered. 

Another point is the running conditions, i.e. the station temperatures. Hostile areas could have very high and very low temperatures, interfering with data collection, wifi communication and data processing. 

References are adequate.

Row 73, the URL link should be written.

Figure 2 is nice but should be deleted. I do not believe a similar drone will be available in 1896 and the Predator in the image will be available since 1940s.

The 4 rows of section 3.3 should be removed.

Authors should add some references on the examples in rows 180-183.

A sequence diagram will help to show the sequence of steps in data communication between station and drone.

What is the measurement unit of scales in images 9 and 10?

Round 2

Reviewer 3 Report

Authors addressed almost all points.

The rows added at end of section 5 describe future works and not one user scenario. It should be a concrete situation where the proposed solution can be applied having relevant advantages. Just two examples: erosion in canyon or glacier status. What sensors could be used and what hostile environment problems will be encountered and addressed.

Author Response

Included the following text at the end of section 5.

As an example of possible scenarios for using data collection in IoT Off-Grid environments, it is necessary to consider, among others, environmental disasters, monitoring of rural areas, monitoring of forests, monitoring of erosion, and hydrological monitoring.

Among others, the sensors that can be used are temperature, humidity, movement, and gas.

Due to possible severe weather conditions in these environments, special attention must be paid during the construction of these modules to withstand such weather conditions.